# Harnessing Skyrmion Hall Effect by Thickness Gradients in Wedge-Shaped Samples of Cubic Helimagnets

**DOI:** 10.3390/nano13142073

**Published:** 2023-07-14

**Authors:** Takayuki Shigenaga, Andrey O. Leonov

**Affiliations:** 1International Institute for Sustainability with Knotted Chiral Meta Matter, Kagamiyama, Higashihiroshima 739-8511, Hiroshima, Japan; 2Department of Chemistry, Faculty of Science, Hiroshima University Kagamiyama, Higashihiroshima 739-8526, Hiroshima, Japan

**Keywords:** skyrmion, surface twists, chiral magnets, FeGe, Dzyaloshinskii–Moriya interaction, conical spiral, skyrmion Hall effect, 75.30.Kz, 12.39.Dc, 75.70.-i

## Abstract

The skyrmion Hall effect, which is regarded as a significant hurdle for skyrmion implementation in thin-film racetrack devices, is theoretically shown to be suppressed in wedge-shaped nanostructures of cubic helimagnets. Under an applied electric current, ordinary isolated skyrmions with the topological charge 1 were found to move along the straight trajectories parallel to the wedge boundaries. Depending on the current density, such skyrmion tracks are located at different thicknesses uphill along the wedge. Numerical simulations show that such an equilibrium is achieved due to the balance between the Magnus force, which instigates skyrmion shift towards the wedge elevation, and the force, which restores the skyrmion position near the sharp wedge boundary due to the minimum of the edge–skyrmion interaction potential. Current-driven dynamics is found to be highly non-linear and to rest on the internal properties of isolated skyrmions in wedge geometries; both the skyrmion size and the helicity are modified in a non-trivial way with an increasing sample thickness. In addition, we supplement the well-known theoretical phase diagram of states in thin layers of chiral magnets with new characteristic lines; in particular, we demonstrate the second-order phase transition between the helical and conical phases with mutually perpendicular wave vectors. Our results are useful from both the fundamental point of view, since they systematize the internal properties of isolated skyrmions, and from the point of view of applications, since they point to the parameter region, where the skyrmion dynamics could be utilized.

## 1. Introduction

Magnetic skyrmions [1,2] are topologically nontrivial spin textures with smooth rotation from magnetization, which are commonly embedded into homogeneously magnetized “parental” states and are stabilized by a specific Dzyaloshinskii–Moriya interaction (DMI) [3,4,5]. DMI performs the function of protecting skyrmions from radial instability and overcoming the constraints of the Hobart–Derrick theorem [6,7]. Phenomenologically, DMI is expressed by the energy terms linear in the first derivatives ofmagnetization M with respect to spatial coordinates, which are so-called Lifshitz invariants (LI):(1)$$L_{i,j}^{\left(k\right)}=M_{i}\partial M_{j}/\partial x_{k}-M_{j}\partial M_{i}/\partial x_{k}.$$
For cubic helimagnets belonging to 23 (T) symmetry classes (as MnSi [8], FeGe [9], and other B20 compounds), which are the subject of the present paper, the Dzyaloshinskii–Moriya interaction reduces to a particularly simple form:(2)$$W_{D}=D\;\left(L_{yx}^{\left(z\right)}+L_{xz}^{\left(y\right)}+L_{zy}^{\left(x\right)}\right)=D\;M\cdot \mathrm{rot}M.$$
and underlies the Bloch-like fashion of the magnetization rotation with the helicity value, γ=π/2 [1].

The characteristic length scale of isolated skyrmions (IS) [10,11] is specified by the competition between ferromagnetic exchange and DM interactions [3,4,5] and ranges from nanometers to microns [12]. In a cubic helimagnet FeGe, the periodicity length λ of the helix in zero field was found to be 70 nm (or 149 unit cells) [13]. Cubic B20 alloys Fe${}_{x}$Co${}_{1-x}$Si have an even larger period (with a maximal value of 230 nm for *x* = 0.3) [14,15]. Topologically, skyrmions are elements of the second homotopy group [16] $\pi_{2}\left(S^{2}\right)=Z$ and are characterized by integer-valued topological invariants, the skyrmion numbers *Q* [2,16], which also indicate that skyrmions cannot be continuously unwound into a homogeneous state without introducing point defects [17]. $Q=\frac{1}{4\pi }\int m\left(\frac{\partial m}{\partial x}\times \frac{\partial m}{\partial y}\right)dxdy$.

During the first years of skyrmionics [8,18] an intensive search for chiral skyrmions was undertaken in bulk samples of cubic helimagnets such as FeGe [9,13,18]. Experimental results clearly indicated a complex multidimensional character of chiral modulated states in small regions of the phase diagrams near the ordering temperatures (so-called precursor states [18,19]). Subsequently, chiral skyrmions (isolated and bound into hexagonal lattices) have been microscopically observed in thin layers of cubic helimagnets (Fe,Co)Si [15] and FeGe [20] in a broad range of temperatures and magnetic fields far below the precursor regions.

The enhanced stability skyrmions’ gain in thin layers of chiral magnets rests on additional surface twists [21,22] of the magnetization near the confining surfaces. Indeed, in bulk helimagnets, only the Lifshitz invariants $L_{x,y}^{\left(x,y\right)}$ in (Equation 1) induce the magnetization rotation in the transverse plane xy. In magnetic nanolayers, on the contrary, the Lifshitz invariants $L_{x,y}^{\left(z\right)}$ with the magnetization derivative along *z* play a decisive role. This is the same energy term that also stipulates the magnetization rotation within the conical phase with the wave vector q along *z*. For skyrmions, as a result, $L_{x,y}^{\left(z\right)}$ leads to the gradual variation in the skyrmion helicity (γ=π/2±δ(z)) towards upper and lower surfaces [21,22]. This effect accumulates additional negative energy as compared with the cones and leads to the SkL stability [22,23]. Such an effect of confinement is mostly manifested when the size of the system is comparable with the characteristic length corresponding to the order parameters of the chiral medium. In particular, the SkL pocket at the phase diagram (film thickness, *T*—an applied magnetic field, *H*) extends up to the confinement ratio ν=T/λ≈8, where λ is a spiral pitch [22,23].

In modern skyrmionics, skyrmions are considered as promising objects for the next-generation memory and logic devices. Indeed, they are often hailed as being “topologically protected” [24], they have a nanometer size [12] and can be manipulated by electric currents [25,26,27]. In particular, in the skyrmion racetrack [28,29], information flow is encoded in isolated skyrmions moving within a narrow strip. At the same time, there is an obvious obstacle in the way of the practical use of skyrmion-based devices—the skyrmion Hall effect (SHE), which leads to the curved trajectory of moving skyrmions due to the transverse skyrmion deflection [30,31]. In turn, this imposes the problem of skyrmion confinement within the nanostrip, which is usually achieved by the edge states of the magnetization rotating towards the free boundaries. The main strategy to overcome this obstacle is to consider localized skyrmion particles with zero topological charge, which would be able to cancel the Magnus force. Among such skyrmion varieties are skyrmionium [32] (also called target-skyrmion [33]), antiferromagnetic skyrmion [34], and/or states of coupled merons with the opposite topological charges [35].

In the present paper, we introduce an alternative mechanism to harness the skyrmion Hall effect. We consider ordinary isolated skyrmions with the unit topological charge. However, we place them into wedge-shaped samples of cubic helimagnets with gradually increasing thickness. Then, the suppression of the SHE and the control over the skyrmion trajectory is based on the balance of the Magnus force, which pushes skyrmions towards the increasing thickness of the wedge (and thus leads to their energy increase) and the restoring force, which tries to return the skyrmion into a position with the minimal eigen-energy near the thinner wedge boundary. We show that this unique equilibrium, which results in straight skyrmion trajectories parallel to the wedge boundaries, is achieved in a wide range of current densities, but ceases to exist at some critical current value. For this, skyrmions are uncontrollably deflected in the transverse direction, irrespective of their increasing eigen-energy.

We also focus on the internal properties of such skyrmion states, which exhibit drastic variation, e.g., of their helicity and characteristic sizes, depending on the film thickness. We complement the phase diagram of states in thin layers of chiral magnets and point to the parameter region, where the mentioned dynamic mechanism could be utilized. In particular, we show that the phase transition between the helical state with the wave vector perpendicular to the field and the conical state with the wave vector along the field may become of the second-order with a smooth transformation between the two spiral states. Moreover, detailed numerical analysis of the current-induced skyrmion dynamics makes it possible to underlie two distinct regimes with different spacings between skyrmion tracks/trajectories for the same increment of the current density. We argue that augmented internal properties of isolated skyrmions in wedge geometries make them effective candidates to be employed in the nanoelectronic devices of the next generation in which nanopatterning is boiled down to a minimum.

## 2. Phenomenological Model and Equations

In the present paper, we use the simplest model for magnetic states in confined noncentrosymmetric ferromagnets based on the following energy density functional [4,5,36]:(3)$$\varepsilon \left(m\right)=A\;{\left(\mathrm{grad}\;m\right)}^{2}+D\;m\cdot \mathrm{rot}\;m-\mu_{0}\;Mm\cdot H,$$
where A>0 and *D* are coefficients of exchange and Dzyaloshinskii–Moriya interactions; H is a magnetic field, which is applied along the *z*-axis; and $x_{i}$ are the Cartesian components of the spatial variable. In this form, functional (Equation 3) includes only basic interactions essential for stabilizing different modulated and localized states.

As a primary numerical tool to minimize the functional (Equation 3), we use the MuMax3 software package (version 3.10), which calculates magnetization dynamics by solving the Landau–Lifshitz equation using the finite-difference discretization technique (see for details Ref. [37]). MuMax3 is a reliable GPU-accelerated software for addressing samples with complex geometries, including notches [38]. For this, we introduce non-dimensional units for the reduced magnetic field, $h=H/H_{D}$, and spatial coordinates, $x_{i}/\lambda$. In particular, the thickness is expressed as ν=T/λ. Here, $L_{D}=A/\left|D\right|$ is the characteristic length unit of the modulated states. The value $\lambda =4\pi L_{D}$ for H=0 is the *helix period* for bulk helimagnets. $H_{D}=D^{2}/A\left|M\right|$. Different micromagnetic structures are described via the magnetization vector m(x,y,z)=M/|M| with a fixed length normalized to unity.

In the subsequent simulations on the current-induced skyrmion dynamics (Section 4.3), however, we use the material parameters for FeGe: DMI, D=0.85272 mJ/m${}^{2}$; the saturation magnetization, $M_{s}=384\;$ kA/m; and the exchange stiffness, $A=4.75\times 10^{-12}$ J/m [18]. At the same time, we omit the contribution of the cubic anisotropy, which may have some impact on the obtained solutions. In particular, it is known that due to the cubic anisotropy, the propagation directions of the spirals point to <100> crystallographic axes just below the Curie temperature and change to <111> when decreasing the temperature below 220 K [13]. Moreover, the symmetry analysis of the P213 structure of FeGe made by Bak and Jensen [36] and by Nakanishi [39] justified that the expansion of the free energy in a slow-varying spin density M(r), besides gradient terms of Equation (Equation 3), must be supplemented by the gradient term of the exchange anisotropy. This energy term will be, however, also omitted in the present paper, although it is known to induce spiral flips into positions oblique with respect to the field direction [40].

We investigate functional (Equation 3) in thin films with the constant thickness *T* and in wedges with varying thickness *T*. The free boundary conditions are used on these confining upper and lower surfaces. Both geometries are infinite along *y* directions (which is the direction of skyrmion motion and current flow) and are confined by the vertical lateral boundaries with free conditions (which confine skyrmions within the racetracks). The tilt angle α characterizes the wedge slope (α=0 in thin films). For simplicity [and following previous publications [22,23,41] we neglect the demagnetizing effects, although we realize that at small film thicknesses they might play an important role. The effect of magnetostatic interactions will be considered elsewhere as soon as the phase diagram of states in thin-film samples, which is the subject of many publications, is completed without them. Moreover, since the information on the parameter range at which skyrmions in wedges could be utilized is extracted from the corresponding phase diagram for thin films, one needs the same phase diagram for thin films, taking into account the demagnetizing fields. At the moment, we assume that the DM interactions suppress demagnetization effects and are the main driving force leading to the magnetization rotation and to the equilibrium periodicity.

At this point, we also introduce a conical phase—a spiral solution of functional (Equation 3) with the propagation direction along the magnetic field in which the magnetization rotation retains a single-harmonic character. The order parameter can be presented as m=(sinθcosψ(z);sinθsinψ(z);cosθ), and the equilibrium parameters are expressed in analytical form [36] as
(4)$$\psi \left(z\right)=\frac{2\pi z}{\lambda },\;\cos\theta =\frac{2|H|}{H_{D}}.$$
For both bulk and thin-film helimagnets, the saturation field of the conical state in units of $H_{D}$ equals h=0.5 when cosθ=1. At this field, the conical phase transforms into the saturated state with θ=0.

## 3. The Simplified Phase Diagram of States in Thin-Layered Systems

In the previous studies [22,23], it was found that the hexagonal skyrmion lattice (hSkL) occupies a vast area at the phase diagram (ν)–(h); at lower fields, it borders the helical state, and at higher fields, it borders the conical (for larger ν) or the homogeneous state (for smaller ν). In theory, such an hSkL represents an ideal state, the energy density (Equation 3) of which is just minimized with respect to the hexagonal cell size and computed for different values of ν and *h*. In experiment, however (see, e.g., Ref. [22] for details), the SkL nucleates at the thin boundary of a wedge-shaped sample and thus creates an energetically costly domain boundary with respect to the conical state. In general, due to the attracting skyrmion interaction, ISs form clusters being surrounded by the conical phase (see, e.g., Refs. [42,43]). This results in a much smaller critical film thickness ν for SkL/cluster stability presumably comparable with the parameter region, in which the energy of an IS becomes negative with respect to the conical state. Taking into account these experimental scenarios, we first look for the mentioned region of ISs with the negative eigen-energy and omit the region of hSkL stability. Thus, the following states will be included into the phase diagram: helicoids, cones, the homogeneous state and ISs (Figure 1a).

### 3.1. Helicoid–Cone Phase Transition

The helicoid (also called a chiral soliton lattice, CSL [44]) occupies the blue-shaded area of the phase diagram (Figure 1a). The wave vector of CSL lies within the plane xy (for definiteness, along *x*-axis) (Figure 1b). For low field values, such spirals have lower energy compared to the conical state, which is readily explained by the additional surface twists for any thickness of the film (mechanism described in the introduction). This effect accumulates additional negative energy as opposed to the cones [21,22], which are not decorated by the surface twists [21,22]. Cones in this geometry (schematically shown in Figure 1c) are oriented along the field and stabilize in the green-shaded region of the phase diagram (Figure 1a). Since usually the film thickness is a non-integer multiple of the helical wavelength, the magnetization of the cones has an uncompensated value of the in-plane component $m_{⊥}$ even in a zero magnetic field. Along the line *A*–*b* (Figure 1a), the CSL transforms into the conical state by the first-order phase transition, which must be accompanied by the coexisting domains of both phases, which are readily resolved experimentally by Lorentz transmission electron microscopy investigations in thin layers of cubic helimagnets, e.g., in FeGe (see Figure 4 in Ref. [22]). At the line *A*–*c* the CSL period tends towards infinity, as observed experimentally in the helimagnet Cr${}_{1/3}$NbS${}_{2}$ [44]. To the left of the point *A* at the line *A*–*a*, however, the CSL-cone transition becomes of second-order, i.e., both processes occur simultaneously. Indeed, for such small thicknesses, the angle spanned by the magnetization in the conical phase becomes less than 2π (first panel in Figure 1c). As a result, the domains of the conical phase nucleate and broaden directly within the helical state; in this case, the rotating magnetization of the conical phase reproduces the surface twists of the helicoid, as schematically depicted in Figure 1b.

### 3.2. Overlap of Surface Twists in Isolated Skyrmions within Thin-Film Helimagnets

In the red-shaded region, the energy of an IS becomes negative with respect to the surrounding state—with respect to the homogeneous state above the line of the cone saturation, h=0.5 (Equation 4), and with respect to the conical state below the line *B*–*f*. Interestingly, the line *B*–*d* coincides with the expansion process of hSkL, whereas in point *B*, both processes diverge; at line *B*–*e*, which is just an auxiliary line, the hSkL period tends to infinity, and the lattice expands into the homogeneous state (for bulk samples, the field value corresponds to 0.4, as defined in Refs. [10,11]). In the same way, as discussed above for the helical state, one may speculate about the conical phase penetrating the inter-skyrmion space during this realistic scenario.

Remarkably, the whole phase diagram of states constructed in Refs. [22,23], which includes hSkL, can be easily reproduced by the following simplified procedure: one extracts the total energy of the surface twists from an hSkL solution with some large film thickness ν (e.g., ν=3). Then, this energy, which is assumed to be constant for varying film thicknesses, can be added to the solutions for the hSkL in bulk samples without any surface twists. The method, however, ceases to “work” in thin samples. In this case, as shown in Figure 2a for isolated skyrmions, the surface twists start to overlap with the decreasing thickness ν. As a result, the total eigen-energy of ISs exhibits minimum at some critical thickness $\nu_{cr}\approx 0.28$ (Figure 2c). Such a phenomenon prompts the placing of ISs in wedge-shaped geometries, where ISs may find an equilibrium “pit stop” position within the energy minimum (Section 4). At the phase diagram (Figure 1a), the points corresponding to the energy minimum of ISs for different field values are located along the line *C*–*D*. We, however, terminate this line in the stability region of the conical phase at the point *D*, since in this case, ISs undergo instability towards tilted spiral states, which stabilize in the form of a thin “belt” around the line *a*–*A*–*b* (on the properties of tilted spiral states, see for details, e.g., Ref. [41]).

### 3.3. The Properties of ISs in Ultrathin Films

Before we switch to wedge-shaped samples of chiral magnets, which are the main subject of the present paper, we systematize the properties of ISs in thin films with decreasing thickness.

First, we highlight the shape transformation of ISs (Figure 3a–d). We define the thickness-dependent size of ISs, $R_{L}$, according to the commonly used Lilley’s definition [45] as a distance between the points at which the tangents at the inflection points of the skyrmion profiles intersect the lines $m_{z}=1$ (Figure 3b). Additionally, we define the skyrmion size $R_{0}$ at the levels with $m_{z}=0$. Both characteristic radii are shown across the layer interface by blue and red solid lines in Figure 3c. It is seen that ISs, being essentially cylindrical objects for small values of ν (first panel in Figure 3c), take a convex shape (second panel in Figure 3c), which subsequently turns into a concave one (last panel in Figure 3c,d). The critical thickness value, corresponding to this process, $\nu_{0}\approx 0.85$ (Figure 3a), is directly related to the described overlap of surface twists. In Figure 3a, the skyrmion radii are determined at the confining surfaces, $R_{L}^{\mathrm{surf}}$ (red dotted line), and in the middle of the layer, $R_{L}^{\mathrm{middle}}$ (blue solid line).

Second, we notice that the helicity is also subject to thickness change (Figure 3e–h). As shown in Figure 3f,h, the helicity value varies in dependence on the radial coordinate *r* across the IS center. To be consistent in the following speculations, we define IS helicity at the point with the largest in-plane magnetization component, i.e., at the point with $m_{z}=0$. In Figure 3g, we plot this helicity value dependent on the *z* coordinate for samples with different film thicknesses. As shown by the helicity profiles, the magnetization continuously rotates from the upper to the lower surface (dotted profiles in Figure 3g). For larger film thicknesses, the helicity equals π/2 in the middle of the layer (solid lines in Figure 3g), which is the case also for bulk samples. The helicity at the surface, as shown in Figure 3e, may reach very distinct values for thin samples, $\approx 65^{\circ }$, whereas in thicker films, the helicity saturates at the value $\approx 47^{\circ }$ consistent with previous studies [22].

## 4. Static and Dynamic Properties of ISs in Wedge-Shaped Geometries

In the present section, we focus on the modification of skyrmion properties in wedge-shaped samples as compared with their thin-film counterparts. We consider axisymmetric skyrmions within the homogeneous state corresponding to the red-shaded region of the phase diagram (Figure 1a) above the field h=0.5. Partly, this is justified by the feasibility of skyrmion nucleation; since the eigen-energy of ISs is negative in this region, the system facilitates their formation from the sharp end of the wedge specifically with additional notches [46] (see Appendix A). In some sense, such a mechanism of skyrmion nucleation is reminiscent of that in stripes of frustrated magnets described in Ref. [47]. The edge states formed at the lateral boundaries of racetracks in Ref. [47] were found to be topologically non-trivial; under an applied electric current, they emitted and absorbed skyrmions and antiskyrmions. A similar mechanism of skyrmion nucleation in wedge-shaped samples was recently modeled and observed in Ref. [48] as also applied to a frustrated magnet Fe${}_{3}$Sn${}_{2}$. Instead of a spin-polarized current, however, the authors of Ref. [48] induced a reversible transformation from magnetic stripes to skyrmions, and vice versa, by a switching of the in-plane magnetic field. The process of skyrmion nucleation in wedge-shaped helimagnets will be considered in detail elsewhere.

In the following simulations, we also avoid the parameter region in which ISs are surrounded by the conical state below the line h=0.5 (Figure 1a). In Ref. [41], the nascent conical state was shown to shape the skyrmion internal structure in such a way that ISs developed anisotropic inter-skyrmion and edge-skyrmion attraction. Such non-axisymmetric ISs are expected to demonstrate complex dynamic behavior (will also be considered elsewhere).

The schematics of the problem are shown in Figure 4a. The position of the skyrmion center with respect to the sharp boundary of the wedge is characterized by the distance *d*. The current is applied along the *y* direction to ensure that ISs move along the thickness gradient due to the non-adiabatic spin-transfer torque, i.e., ISs “climb” the wedge “hill”. The Landau–Lifshitz–Gilbert equation is thus supplemented by the Zhang-Li spin-transfer torque (see formula (26) in Ref. [37]). Current spin polarization rate, P=0.1; the Gilbert damping, 0.066; nonadiabaticity of spin-transfer torque, ξ=0.1. An applied magnetic field H=0.19932T corresponds to the nondimensional value h=0.5. For dimensional simulations, we use the grid with the cell size 2 nm.

### 4.1. Internal Structure of Edge States in Wedge Geometries

In thin-film magnetic nanostructures, the practical use of skyrmions hinges on their repulsion from lateral edges, which would keep moving skyrmions within the racetracks [49]. In chiral magnets, such a repulsion is naturally provided by the DMI, which tilts the magnetization vector away from the magnetic field direction at the edges of a magnet, giving rise to the so-called edge states [50,51]. The interaction potential between edge states and ISs is “flat”, and according to the model in Ref. [52], it decays exponentially with the skyrmion–edge separation distance, i.e., for zero current, the skyrmions are slowly pushed away from the edges. In Figure 6a, we plot interaction potentials Φ(y) for thin films with different thicknesses ν. Φ(y) were calculated by imposing the constraint $m_{z}=-1$ at the skyrmion center and minimizing the energy with respect to spins at all other sites. Then, Φ represents an energy difference of states with and without ISs, i.e., the skyrmion eigen-energy.

In wedge-shaped nanostructures (Figure 4a), however, the internal structure of edge modulations becomes more involved. Figure 4b shows $m_{x}$ and $m_{z}$ components of the magnetization (first two panels) as well as the energy density (third panel). We zoom the color plots within a particular range to reflect all the subtleties of the magnetization distribution. For example, the $m_{z}$ component is highlighted in the range [0.9,1]. We notice that the main magnetic inhomogeneity is localized near the sharp boundary of the wedge. Just like in the case of thin films, the energy of these edge states is negative as compared with the homogeneous state. The homogeneous state on its own accepts the value $m_{z}=1$ only deep inside the wedge (second panel in Figure 4b), and the magnetization contorts towards the upper confining surface. From the behavior of the $m_{x}$ magnetization component (first panel in Figure 4b), we can deduce that the magnetization twists towards the surface, which leads to the layer of the negative energy density also immediately under the confining surface (third panel in Figure 4b). Interestingly, such layers with a negative energy density are also spotted across the whole wedge, indicating small perturbations of the homogeneous state.

### 4.2. Skyrmion “Pit Stops”

In Figure 4c,d, we plot the eigen-energy of ISs, Φ, depending on the distance *d* from the sharp wedge boundary. The energy density distribution in this case is shown in the inset. The skyrmion core with a positive energy density (over the homogeneous state) is found to be embraced by a distorted ring with a negative energy density. Towards the sharp edge, this ring leans upon the negative energy of edge states; in the opposite direction, the ring weakens and merges with the homogeneous state. We notice that the edge–IS interaction potentials in this case exhibit the minima (Figure 4c) at about the same thickness, irrespective of the wedge angle α, which roughly corresponds to the critical thickness calculated in Figure 2c for thin-film geometries. The minima originate from the balance between the increasing eigen-energy of ISs while moving into the region with high ν and the edge-skyrmion repulsion. For smaller wedge angles (dark-blue line in Figure 4c,d), the edge states almost do not influence the magnetization distribution within ISs, and the IS energy is essentially the same as for the thin-film geometries in Figure 2c. For larger elevation angles, however, the equilibrium skyrmion position is found within the edge states, which results in an increase in their energy. Notice that the equilibrium skyrmion position *d* in Figure 2d moves closer to the sharp edge until the minimum disappears for larger angles α. According to our simulations, there are two critical wedge angles: (i) in the angle range $\alpha \in [0,13.5^{\circ }]$, the energy of an IS remains negative with respect to the surrounding homogeneous state (i.e., we separately calculated the magnetization distribution and its total energy within a wedge without ISs); however, for $\alpha >13.5^{\circ }$ it becomes positive; (ii) for $\alpha >22^{\circ }$, the energy minimum shallows, which means that ISs existing within the nanostructure would inevitably “slide” towards the wedge boundary and annihilate (Appendix A). We dubbed this energy minimum a skyrmion “pit stop” position. Indeed, every time the current would switch off, ISs return to this equilibrium (Appendix A).

### 4.3. Current-Driven Dynamics of ISs in Wedge-Shaped Geometries

Figure 5a shows the offset skyrmion position *d* as calculated with respect to the sharp edge during the current-driven ascent of the wedge. The wedge angle is $\alpha =10^{\circ }$, and the current increment is $1\times 10^{11}$ A/m${}^{2}$. For each current value, ISs are found to reach a straight trajectory parallel to the *y* axis (see also schematics in Figure 5d). It takes around 300 ns to reach an appropriate track, as can be drawn from the time-dependent characteristics d(t) in Figure 5b. Even for relatively small current values (e.g., for $I=1\times 10^{11}$ A/m${}^{2}$), an IS leaves its “pit stop” position corresponding to the energy minimum (as described in Figure 4d) and elevates until the appropriate thickness is reached to ensure a balance between the restoring force (proportional to the volume change of ISs) and the Magnus force (proportional to the volume of the magnetization inhomogeneity). In some current range (i.e., from $2\times 10^{11}$ A/m${}^{2}$ to $5\times 10^{11}$ A/m${}^{2}$), the correlation between the current magnitude and the distance *d* becomes nearly linear, i.e., equal increments of the spin-polarized current result in the equidistant skyrmion tracks. This effect is related to the curvature of the skyrmion potentials in Figure 4d, giving rise to the restoring forces. At larger current magnitudes, however, this behavior becomes essentially non-linear. Small changes in the current lead to larger skyrmion elevations until, at the critical current $I_{cr}=7.6\times 10^{11}$ A/m${}^{2}$, the restoring force cannot balance the skyrmion Hall effect, which pushes ISs uphill until they reach the upper wedge boundary and probably even penetrate through the edge states and annihilate. One cannot engage the Thiele equation [53] to address the considered breakdown of the balanced skyrmion propagation along the straight tracks. Indeed, a core assumption of the Thiele equation is the unchangeability of the internal structure of ISs. That is why the Thiele equation is usually applied to two-dimensional ISs, with the repulsive interaction between them. But once the skyrmions become essentially three-dimensional and modulate their shape and volume while moving (as systematized in Figure 2), Thiele equation becomes invalid [54].

Remarkably, one can get enhanced control not only over the skyrmion ascent for increasing electric currents, but also over their descent; one can return skyrmions to lower elevations/tracks by decreasing the current density. Grey dashed lines in Figure 5a,b demonstrate the feasibility of such a process; first, we displaced an IS from the track with $I=5\times 10^{11}$ A/m${}^{2}$ to the track with $I=7\times 10^{11}$ A/m${}^{2}$ and then sent it back by dropping the current to its initial value.

### 4.4. Comment on the Skyrmion “Pit Stops” in Truncated Wedges

The internal structure of edge states near the thicker wedge boundary with the angle π/2−α (Figure 4a) is almost the same as the magnetization distribution in thin-film geometries (Figure 6b. Indeed, the wedge slope (e.g., $80^{\circ }$ in Figure 4 and Figure 5) deviates only by a small amount from $90^{\circ }$ in thin films. Then, these edge states underlie repulsive potentials [55,56] as already mentioned in Section 4.1 (Figure 6a). However, the film thickness decreasing towards the sharp end of the wedge (related to the decrease of the IS eigen-energy) would lead even to the stronger skyrmion-edge repulsion, i.e., the “flat” interaction potentials for thin layers (Figure 6a) would accept a negative slope downwards. Thus, those isolated skyrmions, which reached the upper lateral boundary of the wedge for $I>I_{cr}$, will necessarily slide into their “pit stop” positions near the sharp end after the current is switched off.

We also notice the magnetic deformation of the edge states in thin films (Figure 6b) as compared with the “infinite” samples (Figure 6c) having only vertical free boundaries and flat edge states, depending only on one spatial coordinate [51]. In thin films, the magnetic inhomogeneity bears a two-dimensional character with condensed energy “lumps” near the horizontal surfaces (Figure 6b). The skyrmion dynamics in thin films with ν=0.5 are demonstrated in the supplement (see Appendix A). As expected, skyrmions can move along the edge [55,56] until the repulsive force from the edge states is unable to balance the SHE. Then, skyrmions collapse at the racetrack boundary.

Our findings in the truncated wedge-like samples also indicate that the edge states formed around the obtuse angles (Figure 6c) are not very effective in maintaining skyrmions within the racetracks. If, for the wedges with a smaller thickness at the lower end (ν=0.5), the ISs still develop an energy minimum; for larger thicknesses (ν=1), this minimum flattens and practically disappears (Figure 6d). One may conclude that such truncated geometries are less efficient, as opposed to the mentioned wedges with the sharp ends. Surprisingly, skyrmions in such truncated wedge-shaped nanostructures were shown to persist even for much higher wedge angles in Ref. [54]. The reason presumably lies in the considered skyrmion lattice state, which had a negative energy density on its own and would not annihilate.

## 5. Conclusions

Using the basic Dzyaloshinskii model for cubic noncentrosymmetric magnets, we thoroughly investigated the internal structure of isolated 3D skyrmions in thin-film and wedge-shaped nanostructures. We demonstrated the variation in skyrmion shape and helicity, depending on the layer thickness and wedge elevation. These skyrmion properties were shown to contribute to non-linear current-driven dynamics. In a wide range of current densities in wedge-shaped samples, isolated skyrmions move only up to some thickness and maintain their straight trajectory parallel to the sharp end of the wedge. Small current increments and decrements enable skyrmion switching from one track to another and thus may encode the information in the skyrmion position along the wedge. We stress that the practical use of skyrmions in wedge-shaped samples relies on the edge–skyrmion interaction, due to which a stable skyrmion position with the minimal eigen-energy is formed near the sharp wedge boundary. Truncated nanowedges, or the wedges with large angles, are claimed to be less effective in keeping skyrmions within the racetrack. Moreover, the positive eigen-energy of isolated skyrmions in this case impedes their nucleation and may even lead to the breakup of skyrmion filaments into bobbers. In essence, our method to overcome the skyrmion Hall effect is based on a simple idea to push skyrmions into a parameter range where they increase their eigen-energy. A similar goal can be achieved by the modulation of material parameters across the racetrack. For example, the voltage-controlled magnetic anisotropy was shown to pin skyrmions in Ref. [28]. Since the skyrmions in the region with the larger anisotropy value increase their eigen-energy, larger critical currents are needed to deepen them and to push through the barrier. Uniaxial anisotropy could also be employed in wedge-geometries to get control over the ISs instead of the applied magnetic field [57].

Last but not least, we complemented the known phase diagram of states in thin layers of chiral magnets with new findings: (i) we demonstrated the second-order phase transition between helical and conical spirals, which have mutually orthogonal wave vectors; (ii) we found a line along which the energy of isolated skyrmions reaches the minimum and thus permits skyrmion use in wedge-shaped nanostructures. We believe that the ideas of the present paper may open new avenues for the development of new spintronics devices based on the dynamics of magnetic skyrmions.

## Figures and Tables

**Figure 1 nanomaterials-13-02073-f001:**
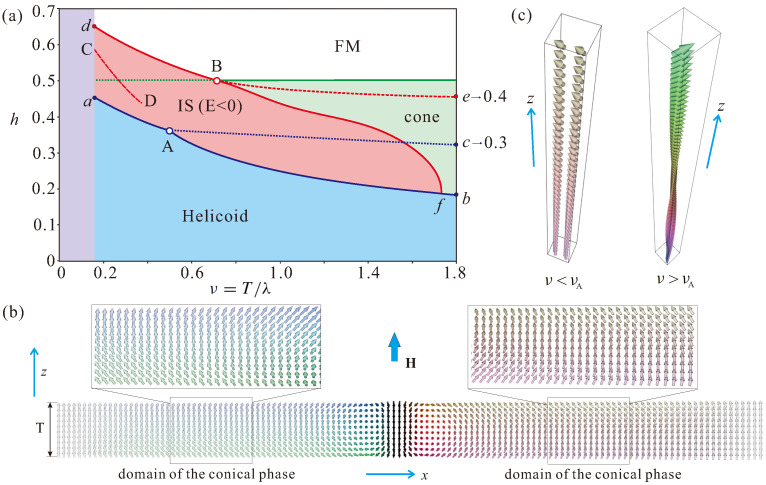
(color online) The simplified phase diagram of states. (**a**) The phase diagram for model (Equation 3) is constructed in reduced variables for the film thickness $\nu =T/4\pi L_{D}$ and the applied magnetic field $h=H/H_{D}$. Filled areas indicate the regions of stability for the helicoid (blue; the wave vector is perpendicular to the field), cone (green; the q-vector is along the field), and the ferromagnetic state. The point *A* is a critical point, in which the phase transition between cones and helicoids changes its type from the first- to the second-order with decreasing layer thickness ν: at the line A−b the first-order phase transition occurs whereas at the line a−A − the second-order one. At the line A−c the period of the spiral state expands to infinity. The line d−B−e is the corresponding process of SkL expansion. In the red-shaded area below the line d−B−f, the eigen-energy of isolated skyrmions becomes negative with respect to the surrounding homogeneous or conical phase. Along the line *C*–*D*, the eigen-energy of skyrmions exhibits minimum for a fixed field value and varying thickness ν. (**b**) Magnetic structure of a helicoid with an expanding period for the field value corresponding to the line *a*–*A*. Comparison with the internal structure of a cone in (**c**) indicates appearance of conical domains with some phase shift formed within the helicoid and thus the second-order nature of their phase transition (see text for details).

**Figure 2 nanomaterials-13-02073-f002:**
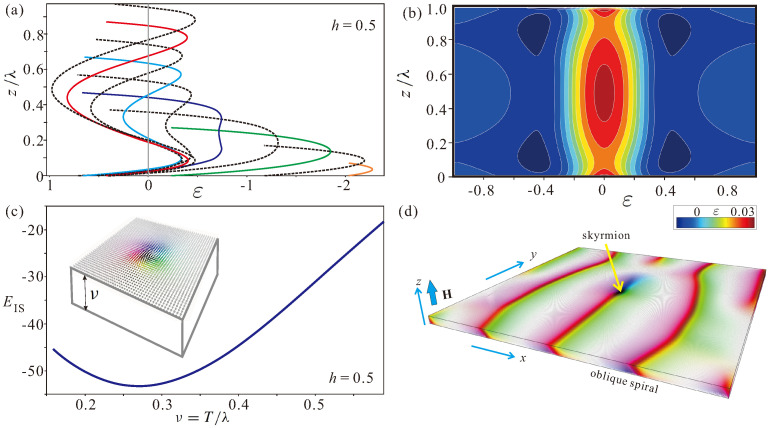
(color online) Overlap of chiral surface twists. (**a**) Energy density ε(z) within isolated skyrmions plotted across the film thickness for different ratios ν/λ. The energy is plotted after integrating over the *x* and *y* coordinates with respect to the energy of the homogeneous state $\varepsilon_{FM}=0.5$. The negative energy near the surfaces is gained due to the additional surface twists of the magnetization. Starting and finishing points of each line correspond to the energy density at the surfaces. In (**b**), the energy density was integrated only with respect to the *y* coordinate and plotted as a color plot on the plane xz. Such an energy distribution exhibits the energy excess in the middle of the layer as well as near the confining surfaces, which may be the underlying reason of toron and bobber formation. Overlap of surface twists in (**a**) results in the thickness-dependent minimum of the total energy of isolated skyrmions (**c**), which also corresponds to the line *C*–*D* at the phase diagram in Figure 1a. To avoid any deformation of the skyrmion shape by the conical phase or an oblique spiral state as depicted in (**d**), we concentrate on the field values h≥0.5 when skyrmions are embraced by the homogeneous state.

**Figure 3 nanomaterials-13-02073-f003:**
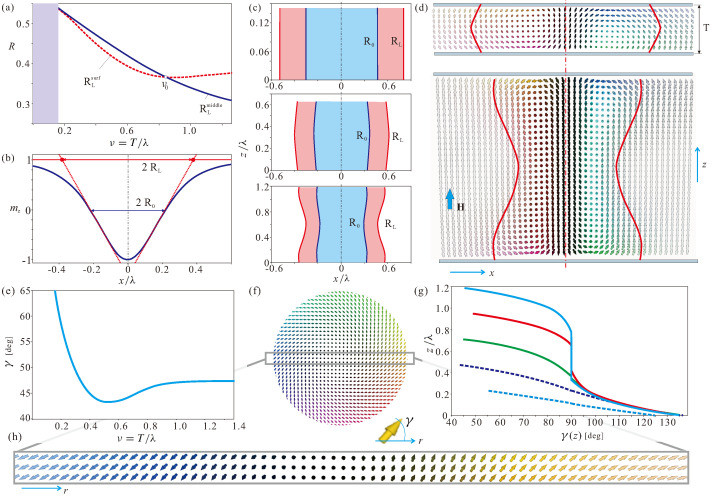
(color online) Variation of skyrmion shapes with decreasing layer thickness ν. The sizes of ISs (**a**) are defined according to the Lilley rule (**b**) by using the lines tangential to the inflection points of the magnetization profiles at the surface and in the middle of the layer. It gives the dependencies $R_{L}^{\mathrm{surf}}\left(\nu \right)$ (dotted red line) and $R_{L}^{\mathrm{middle}}\left(\nu \right)$ (solid blue line in (**a**)) with the crossing point $\nu_{0}$ between them. Profiles $R_{L}\left(z\right)$ across the film interface shown as red lines in (**c**) reveal the skyrmion appearances as convex and/or concave barrels. In addition, we color in blue the skyrmion core with the magnetization rotation from $m_{z}=-1$ to 0. Schematic representation of skyrmions with the two characteristic profiles and the corresponding distribution of the magnetization field are depicted in (**d**). Skyrmion helicity γ is shown to sweep quite a broad angular range for small film thicknesses until it saturates at the value $\approx 47^{\circ }$ (**e**). Interestingly, the helicity value was found to vary across the skyrmion center (**f**) as clearly deduced from the zoomed image (**h**). In (**g**), we plot the helicity profiles across the layer defined under condition $m_{z}=0$ at each coordinate *z*. While the profiles for thicker films have an interval with constant helicity, γ=π/2, which is inherent for bulk cubic helimagnets with the Bloch fashion of the magnetization rotation, in thinner films, the magnetization continuously rotates from the upper to the lower surface (as shown by dotted lines in (**g**)).

**Figure 4 nanomaterials-13-02073-f004:**
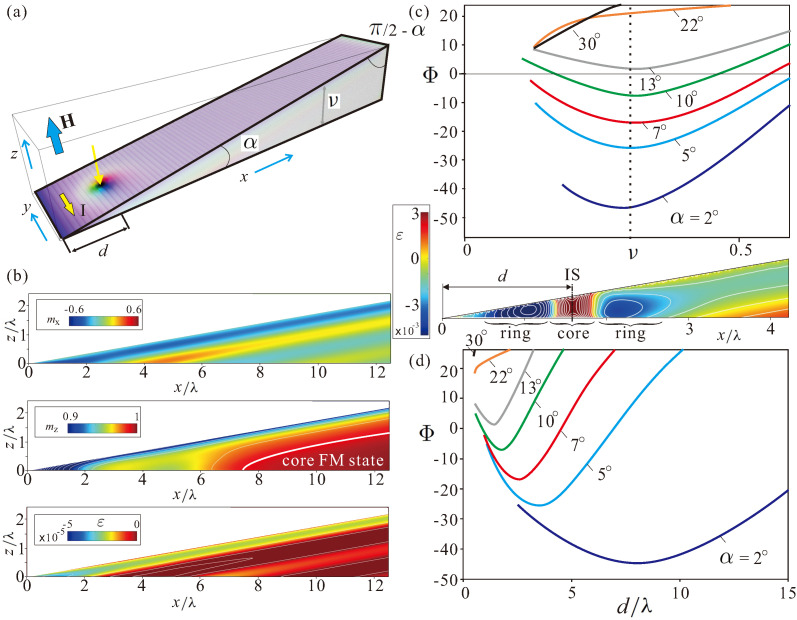
(color online) Static properties of isolated skyrmions in wedge-shape nanostructures. (**a**) Schematic representation of an isolated skyrmion in a wedge with the tilt angle α. The distance *d* of the skyrmion from the sharp edge is measured with respect to the skyrmion center. The corresponding thickness is defined as ν=dtanα. The current *I* is applied along the *y* axis, and the magnetic field is parallel to *z*. As a result, skyrmions are expected to move along the current with a small deviation along *x*. (**b**) The internal structure of so-called edge states arising due to the open boundary conditions. First two panels in (**b**) indicate the rotation of the magnetization towards the upper surface and within the sharp end. The third panel indicates the negative energy density related to these twists. Isolated skyrmions placed within the wedge develop an energy minimum of the edge–IS interaction, which, according to (**c**) is located at about the same skyrmion elevation ν, and according to (**d**), moves closer to the edge. The energy minimum persists only in the angular range $\alpha \in [0,22^{\circ }]$. For larger tilts, the equilibrium “pit stop” position of skyrmions near the edge disappears, and skyrmions, left “alone” within the wedge, would slide down and annihilate. The inset shows the energy density in the wedge, including an IS. A ring of negative energy density known to form around the skyrmion [10] rests upon the part of the negative energy associated with the edge states within the sharp wedge end.

**Figure 5 nanomaterials-13-02073-f005:**
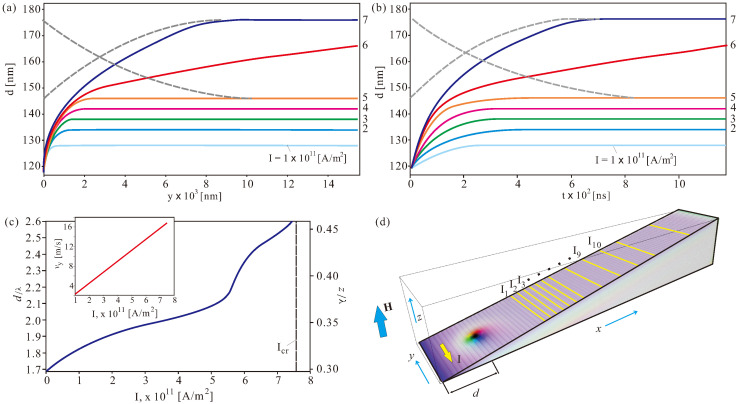
(color online) Current-driven skyrmion dynamics. (**a**) Trajectories of moving isolated skyrmions for different current values indicating both *x* (i.e., *d*) and *y* coordinates of a skyrmion center. After a short time period, when the skyrmion actually “climbs” the hill, the skyrmion track becomes essentially a straight line parallel to the sharp edge. In (**b**), we plot the same trajectories as the time-dependent curves d(t). Panel (**c**) summarizes the information about the skyrmion trajectories; it exhibits the largest skyrmion distance from the edge (left axis) as well as the skyrmion elevation (right axis). The inset shows corresponding velocity along *y*. The current-driven skyrmion motion bears an essentially non-linear character. At smaller magnitudes of the current density, the skyrmion trajectories are almost equidistant, in accordance with the current increment as shown by the yellow lines in (**d**). Larger current densities, however, drive skyrmions to higher elevations, a process which first results in larger separations between skyrmion tracks and eventually culminates in an unbound skyrmion escape up to the upper wedge boundary at the critical current, $I_{cr}\approx 7.6\times 10^{11}$ A/m${}^{2}$. Remarkably, the decreasing current density may induce backward skyrmion movement down the “hill” as shown by the gray dashed lines in (**a**,**b**). During these processes, we first calibrated the skyrmion motion for $I_{1}=5\times 10^{11}$ A/m${}^{2}$. Then, we switched the current to $I_{2}=7\times 10^{11}$ A/m${}^{2}$. As soon as the skyrmion stabilized to its equilibrium motion, we dropped the current back to $5\times 10^{11}$ A/m${}^{2}$, which returned the skyrmion exactly to its initial trajectory.

**Figure 6 nanomaterials-13-02073-f006:**
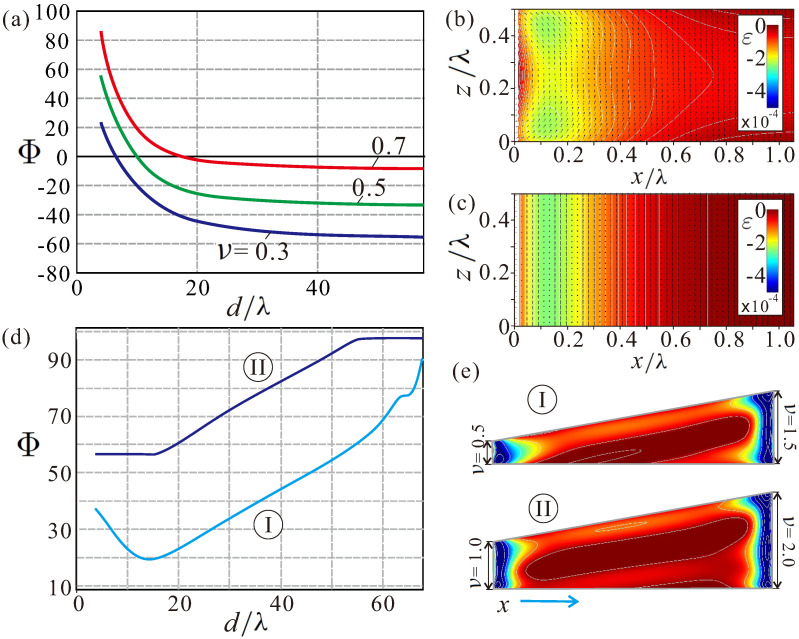
(color online) Static properties of isolated skyrmions in truncated wedge-geometries and thin films. (**a**) Edge-skyrmion interaction potentials in thin films with different thicknesses ν. Note that the curves saturate for the configurations with the ISs located at large distances from the edges. The internal structure of the edge states in thin films (**b**) differs from the essentially one-dimensional edge states in samples with the infinite lateral boundary (**c**). Nevertheless, such a 2D magnetization pattern also underlies the edge-skyrmion repulsion. The edge-skyrmion interaction potential in truncated wedges (**d**) loses its minimum when going from the sample ♯1 with the thinner lower end to the sample ♯2 with the thicker edge. The internal structure of the edge states in truncated wedges (**e**) is similar to those in thin films. However, such edge states are concluded to be less efficient in maintaining skyrmions within the racetracks.

## Data Availability

The data presented in this study are available on request from the corresponding author.

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
