# Peer review of "Harnessing Skyrmion Hall Effect by Thickness Gradients in Wedge-Shaped Samples of Cubic Helimagnets"

_nanomaterials, 2023, doi:10.3390/nano13142073_

Round 1
Reviewer 1 Report
The manuscript is devoted to study, theoretically, a system of a racetrack with varitating thickness in which a skyrmion is driven by spin transfer torque (STT). The objective of the manuscript is to show how this thickness variation can help in supressing the skyrmionic Hall effect.
Overall, the manuscript is well presented and well written. Authors give enough details for understanding and supporting the conclusions. Before recommending publication, however, I think that some points should be clarified:
a) The driven current, that results in STT should be injected in the ferromagnetic sample. The thickness variation would make this current to flow non-uniformly through the sample. As authors state, the effect of the STT is incorporated in the Landau-Lifshitz-Gilbert equation following the standard Zhang-Li torque (ref 36) where uniform current is assumed. Could this be imporant enough? Authors should comment on this, and give some indications on the validity of their assumption.
b) Demagnetizing fields are neglected (or included in the anisotropy), as stated in page 4. This is standard in ultrathin films, but in thickness-dependent samples, this assumptions should be further discussed.
c) Authors state (page 10, line 282) that the edges exerts an exponential force that pushes skyrmion "away" from the edges. Actually, this should be better explained, since the edge produces, also (and for the same reason that there is skyrmionic Hall effect) a component that pushes the skyrmion "along" the edge [see for example NanoLett14, 4432 (2014); PRB94, 184104(2016)].
Author Response
We have attached a pdf-file with the replies to the Referee's comments.

Reviewer 2 Report
The manuscript by Shigenaga et al. presents a theoretical study on skyrmion dynamics in cubic helimagnets. It was shown by micromagnetic simulations that the skyrmion Hall effect can be effectively overcome by exploiting a wedge-shaped track. The physical mechanism for the stable motion of the skyrmion is argued to be a balance achieved between the Magnus force exerted by the driving electric current and the edge-skyrmion interaction due to the thickness gradient. It was also pointed out the position dependence of the moving skyrmion on the current density provides an additional freedom for the manipulation of the skyrmion and in principle an alternating way to code information. I find this work very interesting and valuable to the community. Before I could recommend its publication, I would like to ask the author to address a couple of issues.
First, I think it should be useful to provide some information about the velocity dependence on current density for skyrmions moving in a wedge-shaped sample. Does the shape of the track affect the skyrmion velocity compared to the case of a flat one?
In the case presented in the paper, the Magnus force exerting on skyrmion pushes it to drift to the direction towards the elevation. Would there be an opposite case, i. e., the Magnus force pointing downward?
Author Response

(The authors gave the same response as above.)
